# Correlation of Psychological Factors, Obesity, Serum Cortisol, and C-Reactive Protein in Patients with Fibromyalgia Diagnosed with Obstructive Sleep Apnea and Other Comorbidities

**DOI:** 10.3390/biomedicines12061265

**Published:** 2024-06-06

**Authors:** Edwin Meresh, Kristine Khieu, Jennifer Krupa, McKinney Bull, Miloni Shah, Safiya Aijazi, Drishti Jain, Jade Bae

**Affiliations:** 1Department of Psychiatry & Behavioral Neurosciences, Loyola University Medical Center, 2160 South First Avenue, Maywood, IL 60153, USA; 2Stritch School of Medicine, Loyola University Chicago, 2160 S First Ave, Maywood, IL 60153, USA; kkhieu@luc.edu (K.K.); jkrupa2@luc.edu (J.K.); wbull1@luc.edu (M.B.); mshah24@luc.edu (M.S.); saijazi@luc.edu (S.A.); djain3@luc.edu (D.J.); jbae5@luc.edu (J.B.)

**Keywords:** fibromyalgia, obstructive sleep apnea, comorbidity, obesity, cortisol, depression

## Abstract

Background: Fibromyalgia (FM) is a chronic pain disorder and is associated with disability, and high levels of pain and suffering. FM is known to co-occur with obesity and obstructive sleep apnea (OSA). Individuals with FM often experience symptoms of pain, depression and anxiety, sleep disturbances, and fatigue. These symptoms may be exacerbated by OSA and contribute to the symptoms’ severity in FM. Obesity is a common comorbidity in OSA patients, and as FM and OSA are related in some patients, obesity also may contribute to FM symptom severity. For healthcare providers to effectively manage FM patients, a better understanding of the co-occurrence between these FM comorbidities and psychological factors is needed. Methods: This study was approved by IRB and conducted using a retrospective EPIC chart review. To identify FM, the following ICD-9 codes were used: (729.1) and ICD-10 (M79.7) codes. To identify patients with OSA, the following ICD-9 codes were used: (327.23) and ICD-10 (G47.33). Body Mass Index (BMI), the total number of medical diagnoses, and psychiatric conditions were documented for each patient. The prevalence of psychiatric conditions including depression and anxiety was compared between patients with and without obesity (BMI > 30), and patients with fewer than 25 medical diagnoses and those with 25 or more diagnoses. A chart review was conducted to identify patients with fibromyalgia with prior serum cortisol testing within the last ten years. Cortisol levels were compared and patients were divided into six groups: 1. FM without identified psychiatric conditions; 2. FM with psychiatric diagnosis of adjustment disorders and insomnia; 3. FM with psychiatric diagnosis of depressive disorders; 4. FM with psychiatric diagnosis of bipolar disorders; 5. FM with psychiatric diagnosis of mixed anxiety and depression; 6. FM with psychiatric diagnosis of anxiety disorders. Available C-reactive protein (CRP) values were gathered. Results: The total FM and OSA population was N = 331. The mean age of the patient population was 63.49 years old, with 297 being female. The diagnoses mean was 31.79 ± 17.25 and the mean total psychiatric diagnoses was 2.80 ± 1.66. The mean BMI was 36.69 ± 8.86, with obesity present in 77.95% of the patients. A total of 66.99% of patients had comorbid anxiety and depression with 25 or more medical problems vs. 33.01% of patients who had fewer than 25 medical problems (odds ratio = 1.50). Patients with a BMI < 30 (N = 71) had rates of anxiety and depression at 64.79% and a mean total of 2.79 ± 1.66 psychiatric diagnoses, whereas patients with a BMI > 30 (N = 258) had rates of anxiety and depression at 61.63% (odds ratio = 1.28) and a mean total of 2.80 ± 1.66 psychiatric diagnoses. The most common other psychiatric conditions among FM/OSA patients included hypersomnia and substance use disorders. Cortisol data: Available cortisol results: FM n = 64, female: 59, male: 5, mean age: 63, average BMI: 38.8. The averages for serum cortisol alone for groups 1–6, respectively, are 9.06, 5.49, 13.00, 14.17, 12.25, and 16.03 μg/dL. These results indicate a relatively upward cortisol serum value by the addition of several psychiatric conditions, with the most notable being anxiety for patients with FM. CRP values were available for 53 patients with an average CRP of 4.14. Discussion: Higher rates of anxiety and depression were present in FM patients with 25 or more diagnoses. The odds ratios indicate that a patient with 25 or more medical problems was 1.5 times more likely to have anxiety and depression than those with fewer diagnoses. Additionally, those with a BMI > 30 were 1.3 times more likely to have anxiety and depression than those with a normal BMI. Conclusion: addressing psychological factors in FM and OSA is important as high healthcare utilization is common in patients with FM and OSA.

## 1. Introduction

Fibromyalgia (FM) is a complex pain dysregulation disorder with many associated conditions. These conditions include sleep disturbance, fatigue, and psychological symptoms. The presentation for a patient with fibromyalgia can be heterogeneous, sometimes mimicking other diseases or exacerbated by comorbid conditions. Its diagnostic complexity requires close collaboration among primary care physicians and specialists to navigate broad differentials effectively. Treatment of fibromyalgia is controversial because by definition there are no abnormalities found in diagnostic workups including labs or routine imaging [1]. The organic etiology of the condition has been questioned, with some arguing that fibromyalgia is a purely psychosomatic disorder. Regardless of the underlying basis, patients with fibromyalgia can experience extreme dysfunction, unemployment, and decreased quality of life, especially when compounded by comorbidities like obstructive sleep apnea (OSA). FM may also be a marker for OSA [2,3,4].

Therefore, patients with this disorder can pose a significant healthcare burden as they navigate multiple specialties to provide symptom relief. Epidemiological studies have been limited by a particular location or gender, but some studies suggest the prevalence to be approximately between 2 and 8% [5]. Although using a relatively small population, one study suggests that these patients comprise over 40% of referrals to tertiary pain clinics [6]. This is just one example of the large healthcare utilization associated with fibromyalgia. A broader snapshot would include the many specialists these patients are routinely referred to including but not limited to rheumatologists, psychiatrists, and sleep medicine doctors. Primary care physicians are expected to juggle this complex landscape, with one study reporting that it took approximately 3 years for a patient to receive a diagnosis of fibromyalgia, and over half of the patients studied received this diagnosis in a secondary specialty clinic [7]. The annual direct cost of a patient with fibromyalgia is approximately over $7000 in the United States. This is worsened by indirect costs, which are estimated to be over $6000 [8]. Higher healthcare costs are associated with a lack of diagnosis [7]. It is theorized that early diagnosis can lead to better support by multiple medical specialties, with targeted treatment to improve function. By understanding the relationship between obesity, depression, anxiety, and patients with fibromyalgia and OSA, healthcare providers can more effectively manage the health of patients with fibromyalgia.

### 1.1. Fibromyalgia and Obesity

According to the Center for Disease Control and Protection from 2017 to 2020 the prevalence of obesity in the United States was 41.9%, and that number continues to rise [9]. It is important to further contextualize fibromyalgia with this increasingly prevalent patient population. While the etiology of fibromyalgia remains unclear, emerging evidence suggests that obesity may exacerbate its symptoms. Delayed diagnosis is associated with obesity and delayed diagnosis is also associated with increased symptom severity. Dysregulation in neuroendocrine function in patients with obesity may contribute and worsen the complex symptomatology seen in fibromyalgia [10]. Multiple studies have indicated that a higher BMI is associated with more severe pain symptoms and reduced quality of life among individuals with fibromyalgia [11,12,13]. Additionally, research suggests a correlation between elevated BMI and cognitive dysfunction, particularly in attention and memory, in this patient population [14]. Increased levels of cortisol have been theorized to explain this phenomenon; however, more research is required in this area [15]. 

### 1.2. Fibromyalgia and Cortisol

FM may be related to hyperactivity of the hypothalamic–pituitary-adrenal (HPA) axis, with one study showing increased cortisol upon waking in 20 patients with FM compared with controls [16], but there has yet to be a meta-analysis that shows significant changes in serum levels of cortisol in these patients [17]. Patients with FM have a disrupted HPA axis in which hyposecretion of cortisol may be present [18]. FM patients have comorbid depression, anxiety, and obesity [19]. The HPA axis for depression is hypercortisolemia, which is linked to a worsened quality of life [20]. The stress axis in FM may be exacerbated by the presence of depression, anxiety, sleep problems, endocrine disturbances, and pain, but the relationship of these factors and the cortisol link is unclear [21]. FM is not a homogeneous diagnosis, and patients may receive a depression diagnosis and often have a conflicting serum cortisol level discrepancy. Patients that have no depression symptoms have even further lower cortisol symptoms with secondary chronic fatigue syndrome. It has been found that a chronically stressed HPA can lead to a decreased production of cortisol over time. This low cortisol may be linked to the musculoskeletal pain and overtiredness seen in chronic pain syndrome. Patients with FM are highly interesting because they display very low cortisol, however, have high depression scores and sleep pattern disturbance [21,22,23]. It is unclear if FM symptoms can lead to comorbid depression or if depression is the causal factor of chronic pain, which further stresses the HPA axis resulting in the hyposecretion of cortisol over time. 

Studies report correlations of FM and cortisol [21,22,23], OSA and cortisol [24,25], depression and cortisol [26,27], and obesity and cortisol [28,29].

### 1.3. Fibromyalgia and Obstructive Sleep Apnea

#### Intersection of FM, OSA, and Obesity

The link between OSA and obesity has been well-documented. Losing weight can treat OSA and lead to a better quality of life, reduce the risk of other correlated conditions such as cardiovascular disease, and can decrease healthcare utilization. A key component of FM is sleep disturbance and a common sleep disorder diagnosed is OSA [30,31]. Previous studies show increased rates of OSA in this population [31]. OSA may contribute to the severity of FM symptoms [19,32]. Untreated sleep symptoms in FM patients worsen feelings of fatigue, severity of pain and mood, and general quality of life [33,34].

### 1.4. Fibromyalgia and Mental Health

Patients with FM often exhibit higher rates of psychiatric disorders, particularly depression and anxiety. A systematic review revealed a weighted prevalence of 52–63% for lifetime major depressive disorder (MDD) and 4–39% for lifetime anxiety disorder among individuals with FM [35]. Understanding the psychological aspects of this patient population is important as these factors can precipitate or exacerbate fibromyalgia symptoms [36]. Specifically, research indicates that depressive symptoms in women with fibromyalgia can intensify feelings of pain and fatigue, and sleep disturbances, and impair physical functionality [37]. Moreover, depression in patients with multiple comorbidities correlates with increased healthcare utilization, likely contributing to the elevated healthcare costs associated with fibromyalgia [38]. The relationship between psychological factors and fibromyalgia symptoms appears to be bidirectional, with fibromyalgia exacerbating symptoms of depression and anxiety, as demonstrated in a study comparing female patients with fibromyalgia in Spain [39].

### 1.5. Research Aim

The aim of this study was to further explore the relationship between fibromyalgia, obstructive sleep apnea, obesity, and psychological symptoms. Existing literature suggests a relationship among these conditions to varying degrees. With a dataset comprising patients diagnosed with both FM and OSA, we sought to address several key questions. First, what proportion of these patients is considered obese? Does higher BMI within this population correlate with increased numbers of depression and anxiety? Is there a correlation between higher BMI and a larger number of total diagnoses, potentially indicating an increased healthcare burden? We hypothesized that a higher total number of diagnoses in this patient population would correlate with increased BMI and higher rates of depression and anxiety.

The secondary aim of the retrospective chart review is to analyze the neuroendocrinology of patients with FM, OSA, obesity, and psychiatric conditions, and understand the range of cortisol serum levels in patients with FM, OSA, and obesity comorbid with psychiatric conditions including depression and anxiety, and those without identified psychiatric comorbidity

## 2. Materials and Methods

### 2.1. Chart Review

This IRB approved retrospective study was conducted using an EPIC chart review. Patients with FM were identified using ICD-9 (729.1) and ICD-10 (M79.7) codes, while patients with OSA were identified using ICD-9 (327.23) and ICD-10 (G47.33) codes. The total number of medical diagnoses, including any psychiatric conditions, and BMI were documented for each patient based on the diagnoses populated in the EPIC Problem List. The prevalence of psychiatric conditions was compared between patients with fewer than 25 medical diagnoses and those with 25 or more diagnoses, as well as between patients considered normal weight, overweight, and obese. A chart review was conducted to identify FM patients with prior serum cortisol testing within the last ten years. The reference value for cortisol serum is 2.9–19.4 μg/dL. Data on psychiatric comorbidities as identified in the chart were gathered. Cortisol levels were compared and the patients were divided into six groups: 1. FM without identified psychiatric conditions; 2. FM with psychiatric diagnosis of adjustment disorders and insomnia; 3. FM with psychiatric diagnosis of depressive disorders; 4. FM with psychiatric diagnosis of bipolar disorders; 5. FM with psychiatric diagnosis of mixed anxiety and depression; 6. FM with psychiatric diagnosis of anxiety disorders. Available C-reactive protein (CRP) values were gathered. 

### 2.2. Obesity Classification

In this study, we evaluated patients who were of normal weight, overweight, and obese, as classified by the National Institute of Health guidelines on BMI. Each patient’s BMI was calculated using their weight in kilograms divided by the square of their height in meters. Table 1 shows the breakdown of BMI for each category.

### 2.3. Data Analysis

We performed descriptive statistical analyses to summarize the demographic characteristics of the patient population, including means and percentages. Additionally, we utilized inferential statistics to examine relationships and differences between variables, conducting a correlation analysis to assess the relationship between BMI and the total number of diagnoses. Subgroup analyses were also conducted to investigate potential differences among subgroups of patients based on their total number of diagnoses (more than or equal to 25 vs. fewer than 25) and weight categories.

## 3. Results

The results and a summary of the patient demographics are in Table 2. The total FM and OSA patient population analyzed was N = 331. The mean age of the patient population was 63.49 years old, with 297 being female and 34 male. The mean BMI was 36.7 ± 8.9, with 78% of the patients being considered obese (BMI > 30) (Table 1). A total of 7% of the patients were of normal weight (BMI between 18.5 and 24.9) and 15% of the patients were considered overweight (BMI between 25 and 29.9). See Figure 1. (Note: one patient was considered underweight, BMI 18.25, and was not included in Figure 1). The mean total number of diagnoses was 31.8 ± 17.3 and the mean total psychiatric diagnoses was 2.8 ± 1.7. The data were also separated into patients with fewer than 25 total diagnoses (N = 123) and patients with more than or equal to 25 total diagnoses (208), as determined by the EPIC Problem List (Figure 2). Of patients with 25 or more medical problems, 66% of patients had comorbid anxiety and depression vs. 55% of patients who had fewer than 25 medical problems (Figure 3). In comparison, patients considered normal weight, overweight, and obese had similar rates of comorbid anxiety and depression, with the percentages being 63% vs. 65% vs. 62%, respectively (Figure 4). Figure 5 shows a scatter plot of all patients and compares BMI to the total number of diagnoses. The correlation coefficient was −0.01, indicating no association between the two variables. This is further confirmed by the bar graph in Figure 4, which shows no statistically significant difference between each group. In addition, no significant difference in the total number of psychiatric diagnoses were seen between each group, both when comparing the total number of diagnoses and BMI, with the average number of psychiatric diagnoses being 2.8 ± 1.7. In addition to depression and anxiety, the most common psychiatric conditions among FM and OSA patients included hypersomnia and substance use disorders. 

Cortisol data: Available cortisol results: FM n = 64, female: 59, male: 5, mean age: 63, average BMI: 38.8. Group 1. FM without identified psychiatric conditions; 2. FM with psychiatric diagnosis of adjustment disorders and insomnia; 3. FM with psychiatric diagnosis of depressive disorders; 4. FM with psychiatric diagnosis of bipolar disorders; 5. FM with psychiatric diagnosis of mixed anxiety and depression; 6. FM with psychiatric diagnosis of anxiety disorders. The averages for serum cortisol alone for groups 1–6, respectively, were 9.06, 5.49, 13.00, 14.17, 12.25, and 16.03 μg/dL (Table 3). These results indicate a relatively upward cortisol serum value by the addition of several psychiatric conditions, most notable being anxiety for patients with FM. 

## 4. Discussion

The findings of this study reveal notable distinctions in psychiatric symptomatology among patients with FM and OSA, particularly with the total number of diagnoses. Patients presenting with 25 or more diagnoses exhibited significantly higher rates of anxiety and depression (66%) compared to those with fewer than 25 diagnoses (55%), suggesting a potential association between disease burden and psychiatric comorbidities. These findings align with previous research indicating a complex relationship between the severity of FM symptoms and the presence of psychological symptoms [35].

Our study did not uncover significant differences in rates of anxiety, depression, or total number of diagnoses among patients classified as overweight or obese compared to those of normal weight. This contrasts with previous theoretical concerns regarding the impact of elevated BMI on patients with fibromyalgia, including increased symptom severity and cognitive dysfunction. However, it is worth noting that our analysis focused solely on the presence of psychiatric diagnoses and did not incorporate subjective measures of symptom severity or cognitive function. In this retrospective pilot study, the correlation focused on FM patients comorbid with obstructive sleep apnea. Psychological factors and obesity should be analyzed in a fibromyalgia-alone group (without OSA). Future investigations should consider implementing subjective pain severity assessments and cognitive function tests to better characterize the relationship between elevated BMI and FM symptomatology. It is likely that FM patients inherently have low cortisol to start with and because of chronic pain and fatigue develop depression, which elevates the cortisol level. Patients with no psychiatric conditions or depression displayed a lower cortisol level compared to patients with depression [32,40,41,42,43,44]. Of all listed conditions, anxiety comorbid with FM overtakes the HPA axis, raising cortisol. FM patients have a wide range of inflammatory cortisol serum values based on psychiatric conditions such as depression, bipolar disease, and anxiety. Based on these conditions, anxiety has the largest impact on chronically elevated cortisol in FM. This has implications for future studies on treatments that may focus on the HPA axis.

Studies report that sleep, pain, and pain perception are related [45,46]. This mechanism could explain the comorbidity of FM and OSA; larger prospective studies are needed. Pain perception is also associated with obesity [47,48]. As obesity is associated with FM and OSA, it is then likely that in a subgroup of patients with FM and comorbid OSA and obesity that all these conditions could be related and lead to increased pain perception (Figure 6 and Figure 7). Neurocircuitry dysfunctions have been reported in OSA, obesity, and in FM [49,50,51]. Prospective studies are needed to see if there is a connecting factor for this possible correlation of FM, OSA, and obesity. It is possible that at least in a subgroup of patients with FM, these patients have had comorbid obesity and OSA, and are at risk of pain and sleep difficulties. Depression is a comorbid condition in FM [52]. Depression is associated with OSA [53]. Prospective studies are needed to see if depression increases pain perception in FM and OSA.

## 5. Limitations

While our study did not reveal a significant correlation between BMI and the total number of diagnoses, it is important to recognize potential limitations in our approach. This may be influenced by the limited sample size and the diverse array of symptom presentations within patients with FM. Future investigations with larger cohorts and more comprehensive assessments of disease burden, including the use of patient surveys, could offer deeper insights into the relationship between BMI and the severity of FM.

This study may also be limited by incomplete documentation of psychiatric comorbidities in electronic medical records (EMR) and potential loss due to follow-up. Our reliance on a chart review introduces variability in diagnostic criteria across healthcare providers. Variations in diagnostic practices, with some providers adopting a more fragmented approach to symptomatology rather than consolidating them under a single diagnostic label, could influence the total number of diagnoses recorded on a patient’s chart. Additionally, establishing a causal link between anxiety, depression, and obesity in FM patients with OSA would require a prospective study. That would allow for the investigation of temporal relationships and the ability to control for potential confounding variables, offering more robust evidence of causality.

This is a pilot retrospective study. An analysis also needs to be conducted on patients with FM alone. In this retrospective pilot study, the correlation focused on FM patients comorbid with OSA. Psychological factors and obesity should be analyzed in a fibromyalgia-alone group (without obstructive sleep apnea). To rule out obstructive sleep apnea, FM patients need to be screened and undergo an overnight sleep study as indicated. Obesity is associated with OSA and FM. A larger prospective study is needed to select FM-only patients in whom OSA is ruled out to see if FM-only patients have obesity. 

## 6. Conclusions

The many comorbidities associated with FM can pose a challenge in developing precise diagnostic criteria for this complex disorder. The high healthcare utilization and burden of comorbidities observed within this patient population underscore the importance of recognizing and managing these concurrent conditions in patient care. Moving forward, it is crucial that future research explores the underlying mechanisms linking FM with these comorbid conditions, especially the potentially exacerbating effects of obesity. Understanding these relationships can help with the development of therapeutic approaches aimed at alleviating symptoms and improving patient outcomes.

## Figures and Tables

**Figure 1 biomedicines-12-01265-f001:**
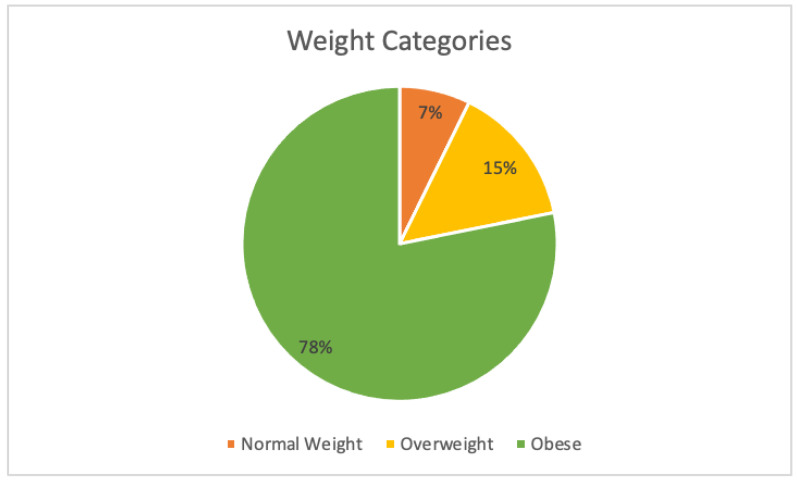
Percentage of patients considered normal weight, overweight, or obese based on BMI. Note: One patient was considered underweight (BMI < 18.5). This patient was not included in Figure 1.

**Figure 2 biomedicines-12-01265-f002:**
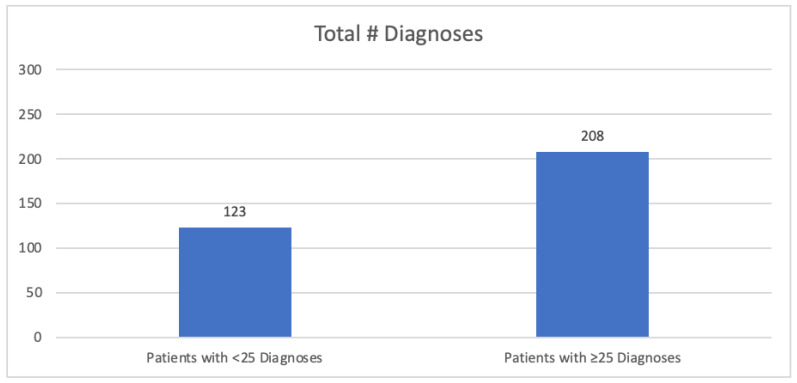
Comparing the total number of diagnoses between patients with fewer than 25 diagnoses and patients with more than or equal to 25 diagnoses.

**Figure 3 biomedicines-12-01265-f003:**
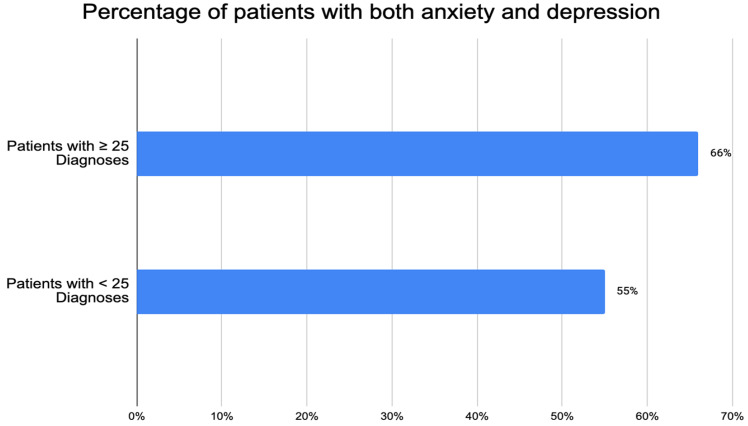
Comparing percentage of patients with both anxiety and depression based on total number of diagnoses.

**Figure 4 biomedicines-12-01265-f004:**
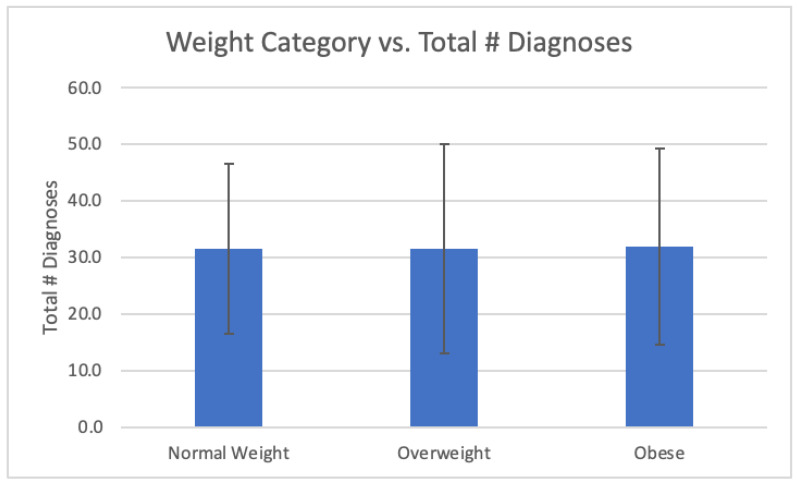
Comparing total number of diagnoses between each weight category.

**Figure 5 biomedicines-12-01265-f005:**
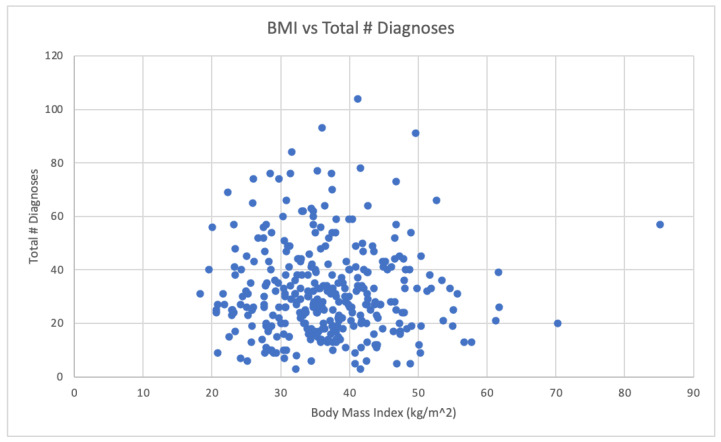
Scatter plot of each patient’s BMI versus their total number of diagnoses. Correlation coefficient = −0.01.

**Figure 6 biomedicines-12-01265-f006:**
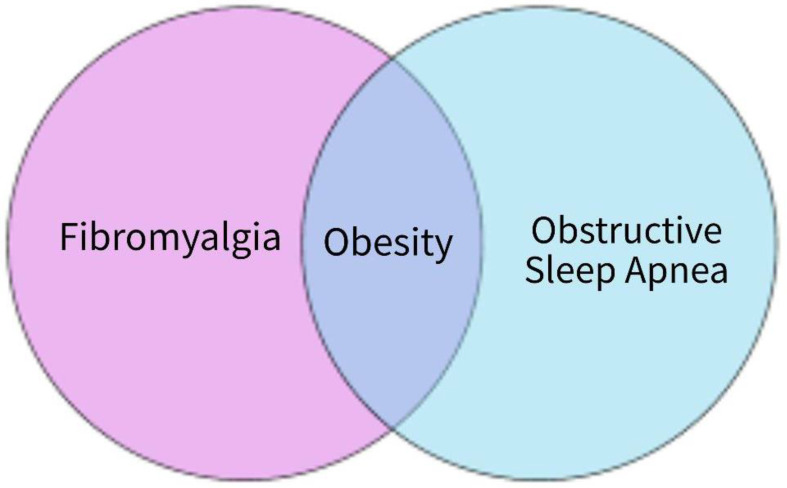
Comorbidity of FM, OSA, and obesity.

**Figure 7 biomedicines-12-01265-f007:**
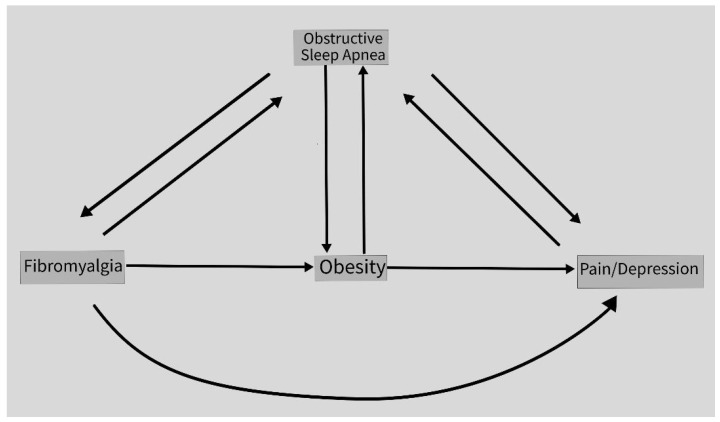
Correlations of FM, OSA, obesity, pain, and depression.

**Table 1 biomedicines-12-01265-t001:** Classification of body mass index.

Category	Body Mass Index (BMI)
Underweight	Less than 18.5
Normal Weight	Between 18.5 and 24.9
Overweight	Between 25 and 29.9
Obese	Greater than 30

**Table 2 biomedicines-12-01265-t002:** Summary of patient demographics and results.

Demographic Characteristics	N	Number of Clinical Diagnoses	BMI	Patients with Anxiety Only	Patients with Depression Only	Patients with Both Anxiety and Depression	Total Psychiatric Diagnoses
Entire Patient Population	331	31.8 ± 17.3	36.7 ± 8.9	26	26	206	2.8 ± 1.7
Patients With <25 Diagnoses	123	16.2 ± 5.4	37.2 ± 8.7	10	14	68	2.4 ± 1.6
Patients With > Or Equal To 25 Diagnoses	208	41.0 ± 15.1	36.4 ± 9.0	16	12	138	3.0 ± 1.7
Normal Weight	24	31.5 ± 15.0	22.6 ± 1.5	1	0	15	2.7 ± 1.3
Overweight	48	31.5 ± 18.5	27.6 ± 1.5	2	3	31	3.3 ± 1.9
Obese	258	31.9 ± 17.3	39.8 ± 7.8	23	23	159	2.7 ± 1.6

**Table 3 biomedicines-12-01265-t003:** FM, psychiatric comorbidities, and serum cortisol.

1: FM without identified psychiatric conditions	cortisol mean: 9.06 μg/dL
2. FM with psychiatric diagnosis of adjustment disorders and insomnia	cortisol mean: 5.49 μg/dL
3. FM with psychiatric diagnosis of depressive disorders	cortisol mean: 13.00 μg/dL
4. FM with psychiatric diagnosis of bipolar disorders	cortisol mean: 14.17 μg/dL
5. FM with psychiatric diagnosis of mixed anxiety and depression	cortisol mean: 12.25 μg/dL
6. FM with psychiatric diagnosis of anxiety disorders.	cortisol mean: 16.03 μg/dL

CRP available for 53 patients, average CRP: 4.14. OSA data: Patients who were diagnosed with OSA had a sleep study completed; 120 had mild OSA and 211 had moderate to severe OSA. The total apnea hypopnea index (AHI) mean was 24.17 and the non-rapid eye movement (NREM) AHI mean was 21.52. CPAP was recommended to patients with moderate and severe OSA (Table 4 and Table 5).

**Table 4 biomedicines-12-01265-t004:** OSA group.

	FM with Mild OSA n = 120	FM with Moderate to Severe OSA n = 211
Conservative management (weight loss)	Yes	Yes
CPAP recommended	No	Yes

**Table 5 biomedicines-12-01265-t005:** AHI in patients with FM and OSA.

FM + OSA		
AHI	24.17731466	NREM AHI mean = 21.5273509

## Data Availability

Data are contained within the article.

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
