# Peer review of "Correlation of Psychological Factors, Obesity, Serum Cortisol, and C-Reactive Protein in Patients with Fibromyalgia Diagnosed with Obstructive Sleep Apnea and Other Comorbidities"

_biomedicines, 2024, doi:10.3390/biomedicines12061265_

Round 1
Reviewer 1 Report
Comments and Suggestions for Authors
To further investigate the relationship between fibromyalgia, obstructive sleep apnea, obesity, and psychological symptoms, this study retrospectively analyzes a data set consisting of patients diagnosed with both fibromyalgia and obstructive sleep apnea.
While this paper is an interesting study, several problems are recognized.
If the purpose of the study was to examine the relationship between fibromyalgia, obstructive sleep apnea, obesity, and psychological symptoms, why did the authors perform their analysis on patients with fibromyalgia and obstructive sleep apnea rather than on those with fibromyalgia alone?
In short, it is well known that obesity is associated with obstructive sleep apnea. Therefore, it is inevitable that patients with fibromyalgia and obstructive sleep apnea will be associated with high BMI and obesity. It seems that an analysis should be performed in patients with fibromyalgia only.
Although the authors state that the purpose of the study was to examine the relationship between fibromyalgia, obstructive sleep apnea, obesity, and psychological symptoms, the discussion in the abstract does not reflect this research purpose.
Author Response
"Please see the attachment."

Reviewer 2 Report
Comments and Suggestions for Authors
Dear authors, I read the article and I have some recomandation
1. paragraf 1.2 we have axis write by 2 times
2. the second paragraph at 1.2 have the same introduction, with the same mistake
3. you write Depression, Obesity with caps lock in the middle of text. please change
4.whats mean "Anxiety ....", what do you want to tell us?
5. the correct form is secondary no secondarily
6. you don't finish the text with point. Why?
7. at the bibliography in the text, sometimes you don't let space, you don't put point, or you put point before the number of the bibliographic titles, please check again
8.If you put the diagnostic of OSA, how many episode of apneea have the patients? They need CPAC? How many hours they use CPAP?
9. it is necessary to make some graphics to show us the relationshion between obesity, OSA, fibromyalgia.
Comments on the Quality of English Language
Please read the text again. Not all the sentences are corect.
Author Response
"Please see the attachment."

Round 2
Reviewer 1 Report
Comments and Suggestions for Authors
no further comments.
Reviewer 2 Report
Comments and Suggestions for Authors
Dear authors, I like the modifications and now the article can be published